# De Novo Assessment and Review of Pan-American Pit Viper Anticoagulant and Procoagulant Venom Activities via Kinetomic Analyses

**DOI:** 10.3390/toxins11020094

**Published:** 2019-02-06

**Authors:** Vance G. Nielsen, Nathaniel Frank, Sam Afshar

**Affiliations:** 1Department of Anesthesiology, University of Arizona College of Medicine, Tucson, AZ 85719, USA; samafshar2013@gmail.com; 2Mtoxins, 1111 Washington Ave, Oshkosh, WI 54901, USA; nate@mtoxins.com

**Keywords:** hemotoxic venom, prothrombin activator, thrombin-like activity, fibrinogenolytic activity, carbon monoxide, heme, metheme, thrombelastography

## Abstract

Snakebite with hemotoxic venom continues to be a major source of morbidity and mortality worldwide. Our laboratory has characterized the coagulopathy that occurs in vitro in human plasma via specialized thrombelastographic methods to determine if venoms are predominantly anticoagulant or procoagulant in nature. Further, the exposure of venoms to carbon monoxide (CO) or *O*-phenylhydroxylamine (PHA) modulate putative heme groups attached to key enzymes has also provided mechanistic insight into the multiple different activities contained in one venom. The present investigation used these techniques to characterize fourteen different venoms obtained from snakes from North, Central, and South America. Further, we review and present previous thrombelastographic-based analyses of eighteen other species from the Americas. Venoms were found to be anticoagulant and procoagulant (thrombin-like activity, thrombin-generating activity). All prospectively assessed venom activities were determined to be heme-modulated except two, wherein both CO and its carrier molecule were found to inhibit activity, while PHA did not affect activity (*Bothriechis schlegelii* and *Crotalus organus abyssus*). When divided by continent, North and Central America contained venoms with mostly anticoagulant activities, several thrombin-like activities, with only two thrombin-generating activity containing venoms. In contrast, most venoms with thrombin-generating activity were located in South America, derived from *Bothrops* species. In conclusion, the kinetomic profiles of venoms obtained from thirty-two Pan-American Pit Viper species are presented. It is anticipated that this approach will be utilized to identify clinically relevant hemotoxic venom enzymatic activity and assess the efficacy of locally delivered CO or systemically administered antivenoms.

## 1. Introduction

Envenomation by hemotoxic enzymes continues to be a major cause of morbidity and mortality worldwide [1], and these venoms are often complex, containing a myriad of diverse enzymes (metalloproteinases, serine proteases, phospholipase A_2_), which have been investigated for medicinal and toxinological purposes [2]. Over the past year, our laboratory has assessed a novel approach to modulating hemotoxic venom activity by directly exposing these substances to carboxyheme and metheme forming agents in insolation [3,4,5,6,7,8,9,10,11,12]. These assessments were performed with thrombelastography, and involved several vipers from Africa [3,4,5], Asia [6,7,8,9], and Australia [5,8], as well as some from the Americas [6,7,10,11]. Importantly, by recently demonstrating that heme was bound to an isolated, purified fibrinogenolytic enzyme derived from *Crotalus atrox* venom that was inhibited by carbon monoxide (CO), our hypothesis that snake venom enzymes may be modulated by heme groups was mechanistically supported [12]. The pattern observed thus far has been that CO inhibits whole venom activity within a concentration range of 70–700 µM, whereas metheme induction by *O*-phenylhydroxylamine (PHA) in the 10-30 mM range may inhibit, enhance, or not affect venom hemotoxic effects [3,4,5,6,7,8,9,10,11]. In sum, several in vitro investigations have demonstrated the effects of heme modulation on hemotoxic venom activity in various regions of the world.

In order to generate a more international assessment of these matters, the present work sought to investigate thrombelastographically uncharacterized venoms obtained from vipers located in North, Central, and South America (Pan-American). For convenience, the various species studied are grouped as anticoagulant (Table 1) or procoagulant (Table 2) based on the indicated available literature [13,14,15,16,17,18,19,20,21,22,23,24,25,26,27,28,29,30,31]. In cases where primary literature describing the venom was unavailable, the designation of procoagulant or anticoagulant activity was guided by the Clinical Toxinology Resources website (www.toxinology.com), an internationally recognized site on envenomation that is maintained by the University of Adelaide in Australia. In terms of definitions, *anticoagulant* venom enzymes have been thrombelastographically documented to exert their effects via fibrinogenolytic [3,6,10,12] or phospholipase [32] mechanisms, which have not been differentiated with our methodology yet. *Procoagulant* venoms with *thrombin-like activity* cause rapid onset of coagulation, but with weak clot strength secondary to lack of engagement of Factor XIII (FXIII) [7,11,31]. In contrast, venoms with what we term *thrombin-generating activity* form thrombin from prothrombin and engage FXIII, which is critical for the development of normal clot strength [33]. It is also important to realize that this kinetic approach has the important limitation of providing primarily functional data that do not discern the enzymatic composition of the proteome of any particular venom. As an example, if a venom is found to be procoagulant with thrombin-generating activity, the enzyme(s) responsible may be metalloproteinases, serine proteases, or another class of enzyme; thus, it may be possible to have two different venoms with identical thrombelastographic coagulation kinetic profiles generated by different proteomes. Conversely, similar venoms from closely related species may have subtle or significant coagulation kinetic profile differences that may potentially be explained by analyses of their venom proteomes. In summary, the present work proposed to use this functional, kinetomic approach to further define the similarities and diversity of Pan-American pit viper venoms.

Taking the aforementioned into consideration, the purposes of the present investigation were as follows: (1) thrombelastographically define in human plasma the effects on coagulation kinetics of these various Pan-American snake hemotoxic venom activities; (2) determine if CO and/or PHA directly inhibits these hemotoxic venom effects; and (3) combine these prospectively obtained data with thrombelastographic data collected with Pan-American venoms recently published [10,11,34,35,36] to assess kinetomic patterns based on location.

## 2. Results

Given the number of abbreviations, Table 3 is provided to assist the reader. For clarity, the results of each type of venom will be presented in a separate section. Coagulation kinetic data and the concentration of venom required to generate it are displayed in Table 4 for anticoagulant venoms and in Table 5 for procoagulant venoms.

### 2.1. Anticoagulant Venom Data

Venom that exerts a primarily anticoagulant effect will generally increase the time to maximum rate of thrombus generation (TMRTG, min, a measure of speed of onset of coagulation), decrease the maximum rate of thrombus generation (MRTG, dynes/cm^2^/sec, a measure of the velocity of clot growth), and decrease total thrombus generation (TTG, dynes/cm^2^, a measure of clot strength) values in venom exposed plasma compared with control plasma conditions without venom addition. Not all anticoagulant venoms uniformly affect TMRTG, MRTG, and TTG, but generally compromise at least two of these parameters. If a carboxyheme state inhibits venom activity, then TMRTG, MRTG, and TTG values of plasma with the addition of carbon monoxide releasing molecule-2 (CORM-2) exposed venom will significantly improve towards values observed in control plasma samples. To confirm that CO is responsible for this inhibition, plasma samples with the addition of venom exposed to inactivated CO releasing molecule (iRM) typically have compromised TMRTG, MRTG, and TTG values similar to those observed in plasma samples with the addition of venom alone. Lastly, if venom is modulated by a metheme state, then plasma samples with the addition of venom exposed to PHA would be expected to have TMRTG, MRTG, and TTG values similar to those observed in control plasma samples. The data concerning the seven anticoagulant venoms are displayed in Table 4.

There was a marked range of concentrations of the venoms to affect anticoagulant effects among the seven vipers tested (0.15–2.0 µg/mL). Venom derived from *Crotalus basiliscus, Crotalus cerastes cercobombus, Porthidium nasutum, Sistrurus catenatus edwardsii,* and *Sistrurus miliarius barbourii* demonstrated anticoagulant activity that was capable of being inhibited by CO. In contrast, venom obtained from *Bothriechis schlegelii* and *Crotalus organus abyssus* had their activities inhibited by both CORM-2 and iRM, not allowing the ability to claim that CO inhibited the anticoagulant enzymes involved. Lastly, only *Crotalus cerastes cercobombus* venom activity was inhibited by PHA. In summary, all of the anticoagulant venoms tested could be considered heme modulated by either CO or PHA, except for venom obtained from *Bothriechis schlegelii* and *Crotalus organus abyssus*.

### 2.2. Procoagulant Venom Data

Venom that exerts a primarily procoagulant effect will generally decrease TMRTG. Venoms with thrombin-like activity that do not activate FXIII will decrease MRTG and decrease TTG values in venom exposed plasma compared with control plasma conditions without venom addition. In contrast, venoms with thrombin-generating activity that do activate FXIII will increase MRTG, but may not significantly affect TTG values in venom exposed plasma compared with control plasma conditions without venom addition. The lack of venom mediated effect on TTG compared with samples without venom exposure is explained by the rate of FXIII activation and fibrin polymer crosslinking plateaus kinetically, thus increases in thrombin generation result in decreases in TMRTG and increases in MRTG values [37]. If a venom’s procoagulant activity was inhibited by either CO or PHA, a significant increase in TMRTG values and increase/decrease in MRTG values towards normal values would be expected. The data concerning the seven procoagulant venoms are displayed in Table 5.

Venom derived from *Agkistrodon bilineatus*; *Bothrops colombiensis*; *Bothrops jararaca*, S; *Bothrops jararaca*, SE; and *Bothrops moojeni* demonstrated thrombin-generating activities that were inhibited by both CO and PHA. Venom obtained from *Atropoides olmec* and *Crotalus simus tzabcan* demonstrated thrombin-like activity. *Crotalus simus tzabcan* venom demonstrated classic kinetics, and its activity was inhibited by both CO and PHA. In contrast, *Atropoides olmec* venom demonstrated a kinetic profile consistent with a weak, thrombin-like activity containing venom with perhaps some activation of factor XIII [31,33]. This venom required the greatest concentration of this group to exert its effects, and the onset time of coagulation was significantly decreased as assessed by split point (SP, minutes, defined as the time from the start of the test to the split of the trace of the thrombelastogram, which corresponds to the onset of fibrin formation), but not by changes in TMRTG—a finding not previously encountered by our laboratory. The values for SP (minutes) for the corresponding experimental conditions were as follows: control (8.1 ± 1.3), venom (3.4 ± 1.3), venom exposed to CORM-2 (7.4 ± 0.8), and venom exposed to iRM (3.8 ± 0.4). Venom significantly decreased SP compared with control conditions, venom exposure to CO exposure significantly prolonged SP values, and iRM exposure had no effect on venom mediated diminution of SP values. Plasma with the addition of venom exposed to CO demonstrated significantly increased MRTG and TTG values, whereas exposure to iRM did not affect venom mediated effects on MRTG and TTG. Lastly, PHA exposure also did not affect *Atropoides olmec* venom activity.

### 2.3. Representative Thrombelastographic Traces of Anticoagulant and Procoagulant Venoms

While the data thus far have been represented by parametric thrombelastographic parameters [3,4,5,6,7,8,9,10,11,12,30], the presentation of traditional, nonparametric traces of the data is also used frequently [29] and can visually complement parametric data. With this in mind, in Figure 1 we present one anticoagulant (panel A), one potent thrombin-generating (panel B), one weak thrombin-generating (panel C), and one thrombin-like (panel D) venom activity under three experimental conditions to illustrate this point.

### 2.4. Kinetomic Review of Pan-American Viper Venoms from This and Previous Publications

In order to assess geographical patterns of kinetomic profiles of Pan-American snake venoms characterized in the present work and in all others using thrombelastographic methods [3,6,7,10,11,29,31,34,35,36], Table 6 was created to review the predominant location, enzymatic activity, concentration of venom, and inhibition to heme modulating agents, as subsequently presented.

In general, it appears that the majority of anticoagulant venoms are found in vipers in North and Central America; procoagulant venoms accounting for a little over half of the typical activities in Central America; and thrombin-generating activity containing venoms found in the majority of venoms in South America, secondary to the predominance of *Bothrops* species assessed thus far. There was a great deal of difference in potency of the venoms (µg/mL) within each activity type, which was determined by approximating the same amount of kinetic anticoagulant or procoagulant kinetic change. The only exceptions to this presentation of potency were the venom concentrations of *Bothrops atrox* and *Bothrops neuwiedi*, the venoms of which were assessed in several systems in addition to the thrombelastograph by a different group of investigators [29] who were not using the performance-based approach we have used. With regard to heme modulation, the majority of venoms tested seem to be affected by CO and/or PHA, with the exceptions mentioned in Table 3 and *Agkistrodon contortrix contortrix* in Table 6. It should be noted that the inability to inhibit *Agkistrodon contortrix contortrix* venom may have been secondary to us not utilizing CORM-2 beyond 100 µM in this earlier work [10], and perhaps a concentration of 1 mM CORM-2 may have inhibited this venom as it did the CO resistant venoms found in Table 4. In summary, there is remarkable diversity in venom activity type, potency, and response to heme modulation across the Americas that has likely been driven in part by predominance of particular species flourishing in different locations (e.g., *Agkistrodon* species in North America, *Bothrops* species in South America), different prey animal species, and other environmental factors.

### 2.5. Effects of Residual CORM-2 and PHA on Human Plasma Coagulation Kinetics Following Addition of Pre-exposed Venom

The present investigation exposed venom mixtures in isolation to CORM-2, iRM, and PHA at the indicated concentrations in Table 4 and Table 5. In some cases, the concentration of CORM-2 was as large as 1 mM, while that of PHA was always 30 mM. Because the venom mixture added to the plasma mixture accounted for 1% of the total volume, the subsequent final plasma mixture concentration for CORM-2 was as large as 10 µM, while it was 0.3 mM for PHA. It was determined previously in a similar plasma-based thrombelastographic series of experiments that coagulation kinetics were not affected by a plasma concentration of CORM-2 of 25 µM or by a PHA concentration of 5 mM [38]. Thus, as the CORM-2 and PHA final plasma concentrations used in all of our investigations over the past few years were smaller than the concentrations already shown to not affect coagulation kinetics, we chose not to repeat similar experiments. However, to allay any concerns, we performed experiments with normal control plasma not exposed to anything, plasma with 10 µM CORM-2, and plasma with 0.3 mM PHA (*n* = 6 replicates per condition), and assessed coagulation kinetics thrombelastographically for 30 min following the methods subsequently described.

Normal plasma samples had a TMRTG value of 16.8 ± 3.8 (minutes); plasma samples with CORM-2 addition had a value of 17.4 ± 3.9, which was not significantly different from normal plasma (*p* = 0.809); and plasma samples with PHA addition had a value of 16.0 ± 3.2, which was not significantly different from normal plasma (*p* = 0.698). Further, normal plasma samples had a MRTG value of 2.4 ± 0.6 (dynes/cm^2^/sec); plasma samples with CORM-2 addition had a value of 2.1 ± 0.6, which was not significantly different from normal plasma (*p* = 0.408); and plasma samples with PHA addition had a value of 2.1 ± 0.6, which was not significantly different from normal plasma (*p* = 0.376). Lastly, normal plasma samples had a TTG of 175 ± 15 (dynes/cm^2^); plasma samples with CORM-2 addition had a value of 174 ± 7, which was not significantly different from normal plasma (*p* = 0.887); and plasma samples with PHA addition had a value of 173 ± 21, which was not significantly different from normal plasma (*p* = 0.831). In summary, neither CORM-2 or PHA at the greatest concentrations used in all of our experimentation over the past year significantly affected human plasma coagulation kinetics.

## 3. Discussion

The present work has displayed some of the most distinct coagulation kinetic patterns in response to heme-based modulation seen yet, every bit as diverse as the African, Asian, and Australian venoms analyzed in recent works [3,4,5,6,7,8,9]. These unique (e.g., *Atropoides olmec*) and similar, but different (e.g., *Bothrops* species) coagulation kinetic responses serve as identifying “fingerprints” of these venoms, and we have coined the term “venom kinetomics” to describe these datasets [8,9]. Rather than a static paradigm, this approach is intended to find new ways to describe the novel effects of toxins on coagulation, providing a quantitative and qualitative backdrop upon which new discoveries can be interpreted. Simple concepts, such as relative potency (e.g., µg/mL) of any particular venom to inflict its anticoagulant or procoagulant effect, are self-evident; however, the effects of heme modulation are more complex and, in a few cases, difficult to interpret. For example, two species, *Bothriechis schlegelii* and *Crotalus organus abyssus*, had significant attenuation of their anticoagulant activities after exposure to either CORM-2 or iRM, without any inhibition from PHA exposure. These two venoms are unique, and while there may very well be a heme attached to key enzymes, one cannot prove definitively that this is the case, as there is an equal CO-independent effect with iRM exposure and no effect with metheme formation with PHA. The use of other carbon monoxide releasing molecules (and iRMs) may be needed in such cases to detect heme modulation. There is also great variation in the amount of CO required to inhibit enzyme activity, with up to 700 µM required to inhibit *Bothrops moojeni* and *Crotalus simus tzabcan* venoms based on the amount of CO released by 1 mM CORM-2 [39] as compared with all the other venoms inhibited by only 70 µM CO. There was also variation in the degree of CO mediated inhibition of procoagulant activity between the other three *Bothrops* species venoms tested, and between the two *Bothrops jararaca* venoms, which is likely secondary to their two distinct proteomes [25]. Specifically, *B. jararaca* S venom had 10.5% metalloproteinase and 28.6% serine protease, whereas *B. jararaca* SE had 35.6% metalloproteinase and 13.7% serine protease in their proteomes [25]. *B. jararaca* S venom had more robust activation of coagulation, less CO mediated inhibition of activity, and more PHA mediated inhibition of activity than *B. jararaca* SE venom (Table 4). Importantly, both metalloproteinase and serine protease enzymes have been demonstrated to have thrombin-generating activity [25], so the variation in proteome almost certainly contributed to the differences in their kinetomic profiles. Lastly, exposure to PHA, a metheme forming agent, resulted in inhibition of venom activity, no effect on venom activity, or potential activation of latent enzymatic activity such as the apparent anticoagulant activity seen with *Agkistrodon bilineatus* and *Bothrops colombiensis* venoms. In summary, our selection of Pan-American venoms displayed a remarkable set of unique kinetomic patterns, many not previously documented with this thrombelastographic methodology.

Continuing refinements to this thrombelastographic methodology will include devising ways to further isolate the effects of specific enzyme types that may contribute to the overall apparent activity of a particular venom. Specifically, while metalloproteinases and serine proteases are well established fibrinogenolytic anticoagulant enzymes [2], the impact of other classes of snake enzyme that anticoagulated by completely different mechanisms, such as phospholipase A_2_ (PLA_2_), have not been extensively quantitated by thrombelastography. Just recently, this laboratory characterized the kinetomic features of such a PLA_2_ purified from *Crotalus adamanteus* venom with thrombelastographic methods, and determined that this enzyme was CO-inhibitable [32]. While of interest, it is important not to generalize this finding to all snake venom derived PLA_2_. This finding is critical, given the diverse effects of snake venom PLA_2_, which include myotoxicity and neurotoxicity [2]. Thus, venoms containing metalloproteinases, serine proteases, and PLA_2_ that display anticoagulant activity in vitro may exert this effect with one or all three enzyme classes; further, CO-mediated inhibition of anticoagulant activity may be secondary to inhibition of one or more of these classes of enzyme. Further development of differential inhibition of the various enzyme classes found in snake venom that is compatible with assessing coagulation with human plasma will be required in the future to increase mechanistic insight into these matters.

Our use of heme modulating agents to change enzymatic activity is new to this field, but it has a substantial scientific grounding. Heme has been found to be attached as an assumed post-translational modification of enzymes on a myriad of proteins [38], and CO binds primarily to transition metals such as the Fe^+2^ found in the center of the heme porphyrin ring [40]. In the absence of an attached heme group, it would not be expected that CO would react with other portions of snake venom proteins as it is not a radical [41], and if not bound to a heme attached to these proteins, then CO would be expected to rapidly diffuse away into the plasma sample. With regard to PHA, this compound is the fastest metheme forming agent available [42], converting the central iron group of heme to an Fe^+3^ state. The approach to identify fibrinogen as a heme modulated molecule included the use of CORM-2, PHA, and mass spectroscopy to explain the kinetic behaviors of fibrinogen in a thrombelastographic investigation [43]. The present investigation involved isolated exposure of venom mixtures with carboxyheme and metheme forming agents at concentrations that, when diluted 100-fold, would not be expected to directly affect plasmatic coagulation, based on previous literature [38]. This was verified with the experiments found in Section 2.5 of Results. In further support of a post translationally bound heme group modulating snake venom enzymes, the kinetic modeling of inhibition of fibrinogenolytic activity of crude *Crotalus atrox* venom [6] and the inhibition of isolated *Crotalus adamanteus* venom derived snake venom anticoagulant activity [32] demonstrated a sigmoidal pattern, which is characteristic of heme-based protein modulation. As a practical matter, while the most direct scientific approach, it would be technically time-consuming and likely very expensive to isolate every protein of every venom individually and assess if heme is attached with mass spectroscopy or other heme detecting methods. It may also not be useful from a clinical or scientific standpoint, as one might need to assess dynamically with CO or PHA via thrombelastography if the predominant activity was modulated to address a given coagulopathy or hypothesis. Therefore, we have taken the approach that if isolated venom activity is affected by heme modulating agents, then there is very likely a heme present as the agents involved are not reacting with the enzyme directly, and the heme modulating agents will be so diluted in the plasma sample that no direct effect by them will be observed thrombelastographically.

While outside the scope of the present work, there has been concern expressed by reviewers over the years about the application of CO locally resulting in systemic CO poisoning. To allay these fears, we offer the following scenario for the readership to consider. If a viper inflicts a bite and releases several milliliters of venom into the muscle of a lower extremity, it dissects through the tissue planes and commences toxic effects. Then, within a few minutes, let it be assumed that some combination of rapid and slow CO releasing compounds with a vasoconstrictor in solution is injected near and within the bite site, resulting in as much as 100 mL of tissue to be exposed to up to 1000 µM CO as a pulse. Then, let it be assumed that this injection continues to release up to 100 µM CO per 100 mL of tissue for up to 4 h. The hope would be that the resident venom enzymes would have heme mediated inhibition, with CO bound to heme until a time occurred when either the enzyme was released into the circulation or the CO was released from the heme group. With regard to CO poisoning, if the entire initial burst of CO occurred within the systemic circulation of a 70 kg human being (estimated blood volume approximately 5 L), then the 100,000 µM of CO from the initial injection would be reduced to a concentration of 20 µM within the central circulation. To complete the analogy, a continuous release over 4 h of 10,000 µM of CO from the bite site would add 2 µM of CO per hour for a total of 8 µM of CO. To put these concentrations of CO into perspective, an addition of CO for a final concentration of 100 µM in whole human blood increased carboxyhemoglobin by only 0.6% [44]; further, smoking two tobacco cigarettes results in an increase in carboxyhemoglobin concentration of 5% [45]. Summated together, the data from the preceding scenario and associated literature demonstrate that the injection of a CORM-based solution to a snake bite would have as much danger of inflicting systemic CO poisoning as smoking one tobacco cigarette.

Another misconception that can be raised when investigating crude venoms is that they may contain iron as trace compounds, such as in hemoglobin or free iron from milking a snake, which may bind CO released from CORM-2 or any other CORM used in experimentation. Biochemically, this is not the case. First, free iron in aerobic solution is rapidly oxidized to the Fe^+3^ state, which does not bind with CO. Second, free iron would rapidly react with whatever anion was near it, and would not function as a modulatory entity of any snake venom enzyme. Similarly, free hemoglobin released into the venom from the trauma of milking the snake would be very small in concentration and would not be expected to somehow rapidly bind to the key enzymes without some sort of intermediary step, such as that associated with post-translational modification during enzyme synthesis within the venom gland cells. Taken as a whole, the use of carboxyheme and metheme forming agents to investigate crude venom is reasonable and not subject to the aforementioned misconception.

In conclusion, the present investigation characterized fourteen Pan-American viper venoms with a novel kinetomic method utilizing thrombelastography. This method has been recently used to characterize venoms derived from African, Asian, and Australian snakes [3,4,5,6,7,8,9]. Using this paradigm, venoms are classified as anticoagulant (fibrinogenolytic or phospholipid digesting) or procoagulant (thrombin-generating, thrombin-like activity), and carboxyheme or metheme responsive [3,4,5,6,7,8,9]. The ongoing determination of vulnerability of venoms to heme modulation is of interest, as it has recently been demonstrated that exposure of *Crotalus atrox* venom to CO markedly inhibited its hemotoxic activity in vivo in a rabbit model [46]. It may be possible that local application of CO via releasing molecules may serve as an adjunct therapy prior to antivenom administration. Finally, it is anticipated that our approach will assist in identifying clinically relevant enzymatic activity and allows assessment of the potential efficacy of antivenoms or heme modulating agents to treat envenomation by hemotoxic venoms.

## 4. Materials and Methods

### 4.1. Venoms, Chemicals, and Human Plasma

Lyophilized venoms depicted in Table 1 and Table 2 were obtained from Mtoxins (Oshkosh, WI, USA) (*Agkistrodon bilineatus*, *Atropoides olmec*, *Bothriechis schlegelii*, *Crotalus basiliscus*, *Crotalus simus tzabcan*, *Porthidium nasutum*) or from the National Natural Toxins Research Center at Texas A&M University (Kingsville, TX, USA) (*Bothrops colombiensis*, *Bothrops moojeni*, *Crotalus cerastes cercobombus*, *Crotalus organus abyssus*, *Sistrurus catenatus edwardsii*, *Sistrurus miliarius barbourii*). The only exception to this was the acquisition of *Bothrops jararaca* venoms, which were a generous gift from Professor Juan J. Calvete at the Instituto de Biomedicina de Valencia, CSIC, Spain. Venoms were dissolved into calcium-free phosphate buffered saline (PBS, Sigma-Aldrich, Saint Louis, MO, USA) to a final 50 mg/mL concentration, aliquoted, and maintained at −80 °C. CORM-2 (tricarbonyldichlororuthenium (II) dimer, a CO releasing molecule), dimethyl sulfoxide (DMSO), and *O*-phenylhydroxylamine (PHA) were acquired from Sigma-Aldrich. Finally, pooled normal human plasma (George King Bio-Medical, Overland Park, KS, USA) that was sodium citrate anticoagulated and maintained at −80 °C was used in all subsequently described work.

### 4.2. Thrombelastographic Analyses

The volumes of subsequently described plasmatic and other additives summed to a final volume of 360 µL. Samples were composed of 320 µL of plasma, 16.4 µL of PBS, 20 µL of 200 mM CaCl_2_, and 3.6 µL of PBS or venom mixture, which were pipetted into a disposable cup in a thrombelastograph^®^ hemostasis system (Model 5000, Haemonetics Inc., Braintree, MA, USA) at 37 °C, and then rapidly mixed by moving the cup up against and then away from the plastic pin five times. The following previously described viscoelastic parameters [3,4,5,6,7,8,9,10,11,12] were measured: time to maximum rate of thrombus generation (TMRTG)—the time interval (minutes) observed prior to maximum speed of clot growth; maximum rate of thrombus generation (MRTG)—the maximum velocity of clot growth observed (dynes/cm^2^/second); and total thrombus generation (TTG, dynes/cm^2^)—the final viscoelastic resistance observed after clot formation. For one procoagulant venom, it was necessary to determine the onset of coagulation with the split point (SP, minutes). SP is defined as the time from the start of the test to the split of the trace of the thrombelastogram, which corresponds to the onset of fibrin formation. Data were collected for 15 min with venoms that were procoagulant, whereas venoms with anticoagulant properties had data collection for 30 min.

The initial concentration for all venoms assessed was 1 µg/mL; if the time to commencement and velocity of coagulation was procoagulant, with the onset of coagulation beginning in half the time or less and/or the speed of clot formation proceeding at two-times or greater than plasma without venom addition, this concentration of venom was used. If this did not occur, then the concentration of venom was gradually increased until these conditions were met. If, in contrast, the venom was anticoagulant in nature, the onset of coagulation had to be twice and/or the velocity of clot formation half of plasma without venom addition in order to be acceptable for investigation. However, if coagulation was not detectable, then the concentration of anticoagulant venom was progressively diminished until at least detectable coagulation occurred; this concentration was then used. In sum, this approach permitted comparison of relative potencies of the venoms and allowed the determination of the predominant effect of a specific venom on coagulation.

### 4.3. CO Exposures

As for exposure to CORM-2 to assess the effects of CO on anticoagulant venom activity, the subsequent four experimental conditions were utilized: (1) control condition—no venom, DMSO 1% addition (*v*/*v*) in PBS; (2) venom condition—venom, DMSO 1% addition (*v*/*v*) in PBS; (3) CO condition—venom, CORM-2 1% addition in DMSO (100–1000 µM final concentration); and (4) inactive releasing molecule (iRM) condition—venom, inactivated CORM-2 1% addition in DMSO (100 µM final concentration). CORM-2 was inactivated as noted [5,8]. Venom was added to PBS with the described additions and incubated for 5 min at room temperature, then 3.6 µL of one of these solutions was added to the plasma sample in the plastic cup. Lastly, in the case of procoagulant venoms, the concentrations chosen were predetermined to result in coagulation kinetic parameter values far different from plasma without additives; thus, no statistical comparison with a control condition was made, as has previously been reported [4,5,7,8].

### 4.4. PHA Exposures

Other experiments involved exposure of all venoms to PHA in order to determine if their activities would change when the conditions favored the formation of metheme (Fe^+3^), as previously described [4,5,8,9]. Venom was placed in PBS with the addition of PHA 3% (*v*/*v*, 30 mM final concentration) for 5 min prior to addition to the aforementioned plasma mixture in a thrombelastographic cup. The results obtained from these experiments were compared to the results generated by condition 2 in the CO exposure series of experiments.

### 4.5. Assessment of Residual Concentrations of CORM-2 and PHA on Human Plasma Coagulation Kinetics Following Addition of Pre-exposed Venom

In order to determine if the highest residual concentrations of CORM-2 and PHA affected plasmatic coagulation, the following experiments were performed. Using the same general procedures seen in Section 4.2 to Section 4.4 of Methods, plasma samples of the following compositions were placed in thrombelastographic plastic cups: 320 µL of plasma, 16.4 µL of PBS, 20 µL of 200 mM CaCl2, and 3.6 µL of PBS (as a representative 0 concentration of other additives) or 1 mM CORM-2 or 30 mM PHA. This resulted in final plasma sample concentrations of 10 µM CORM-2 and 0.3 mM PHA. After mixing, sample data was collected for 30 min. All conditions were represented by *n* = 6 replicates.

### 4.6. Statistical Analyses

Data are presented as mean ± SD. Experimental conditions were composed of *n* = 6 replicates per condition, as this provides a statistical power >0.8 with *p* < 0.05 utilizing these techniques [3,4,5,6,7,8,9,10,11]. A statistical program was used for one-way analyses of variance (ANOVA) comparisons between conditions, followed by Holm–Sidak post hoc analysis or unpaired, or two-tailed Student’s *t*-tests as appropriate (SigmaPlot 14, Systat Software, Inc., San Jose, CA, USA). *p* < 0.05 was considered significant.

## Figures and Tables

**Figure 1 toxins-11-00094-f001:**
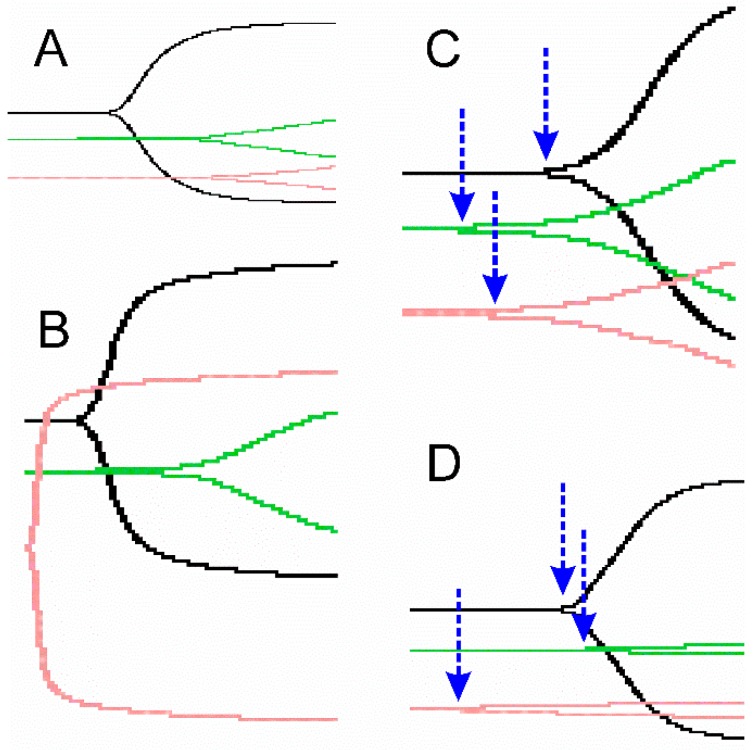
Representative traces of traditional thrombelastograms of four different venoms under three experimental conditions. The pink trace represents venom alone, the black trace represents venom after carbon monoxide (CO) exposure, and the green trace represents venom after *O*-phenylhydroxylamine (PHA) exposure. The blue arrows indicate the split point time for the indicated traces. (**A**) Anticoagulant—*Crotalus basiliscus* traces over 30 min; (**B**) Procoagulant, thrombin-generating activity—*Bothrops moojeni* traces over 15 min; (**C**) Procoagulant, thrombin-like activity—*Atropoides olmec* traces over 15 min; and (**D**) Procoagulant, thrombin-like activity—*Crotalus simus tzabcan* traces over 15 min. The figures have been cropped to maximize the visual effects of the different biochemical conditions and thus should not be considered or compared with the assumption of identical time scales.

**Table 1 toxins-11-00094-t001:** Species of anticoagulant snake venoms investigated.

Species	Common Name	Location
*Bothriechis schlegelii* [13,14]	Eyelash Palm Pit Viper	Central and South America
*Crotalus basiliscus* [15,16]	Mexican West Coast Rattlesnake	West Coast of Mexico
*Crotalus cerastes cercobombus*	Sonoran Desert Sidewinder	Mexico; Southwest Arizona, USA
*Crotalus organus abyssus*	Grand Canyon Rattlesnake	Northern Arizona, USA
*Porthidium nasutum* [17,18]	Rainforest Hognosed Pit Viper	Central and South America
*Sistrurus catenatus edwardsii* [19]	Desert Massasauga	North and Central America
*Sistrurus miliarius barbourii* [20]	Southeastern Pygmy Rattlesnake	Southeastern USA

**Table 2 toxins-11-00094-t002:** Species of procoagulant snake venoms investigated.

Species	Common Name	Location
*Agkistrodon bilineatus* [21]	Common Cantil	El Salvador, Guatemala, Mexico
*Atropoides olmec*	Olmecan Pit Viper	Guatemala, Mexico
*Bothrops colombiensis* [22,23,24]	Common Lancehead	Republic of Columbia
*Bothrops jararaca, S and SE* [25,29]	Jararaca	Argentina, Brazil, Paraguay
*Bothrops moojeni* [26,27,28]	Brazilian Lancehead	Brazil, Paraguay
*Crotalus simus tzabcan* [30,31]	Central American Rattlesnake	Central America

S = southern region of Brazil and SE = southeastern region of Brazil, corresponding to the venomic analyses of these two geographically separated viper populations, as cited in the work of [25].

**Table 3 toxins-11-00094-t003:** Abbreviations.

Abbreviation	Definition
CO	carbon monoxide
CORM-2	tricarbonyldichlororuthenium (II) dimer, a CO releasing molecule
DMSO	dimethyl sulfoxide
FXIII	factor XIII
iRM	inactive releasing molecule
metheme	a heme moiety that has its iron center in the Fe^+3^ rather than the Fe^+2^ state
MRTG	maximum rate of thrombus generation (dynes/cm^2^/second)
PBS	phosphate buffered saline that is calcium free
PHA	*O*-phenylhydroxylamine
TMRTG	time to maximum rate of thrombus generation (minutes)
TTG	total thrombus generation (dynes/cm^2^)
SP	split point, time from the start of the test to the split of the trace (minutes)

**Table 4 toxins-11-00094-t004:** Coagulation kinetics of anticoagulant venoms ± carbon monoxide (CO) or ± *O*-phenylhydroxylamine (PHA) in plasma.

Parameter	Control	V	V/CO	V/iRM	V/PHA
*Bothriechis schlegelii* (0.15 µg/mL)					
TMRTG	11.7 ± 1.2	26.6 ± 2.9 *	14.9 ± 1.7 *†	16.5 ± 1.7 *†	23.1 ± 3.8
MRTG	3.6 ± 0.6	0.8 ± 0.3 *	2.5 ± 0.3 *†	2.0 ± 0.9 *†	1.1 ± 0.4
TTG	203 ± 10	67 ± 32 *	196 ± 14 †	162 ± 66 †	116 ± 56
*Crotalus basiliscus* (1.0 µg/mL)					
TMRTG	11.7 ± 1.2	22.7 ± 4.8 *	13.3 ± 1.1 †	21.4 ± 4.1 *‡	21.8 ± 4.8
MRTG	3.6 ± 0.6	0.3 ± 0.5 *	2.8 ± 0.3 *†	0.9 ± 0.8 *‡	0.6 ± 0.7
TTG	203 ± 10	33 ± 60 *	196 ± 6 †	96 ± 73 *‡	60 ± 73
*Crotalus cerastes cercobombus* (2.0 µg/mL)					
TMRTG	11.8 ± 0.7	19.8 ± 4.0 *	12.5 ± 1.9 †	21.6 ± 4.2 *‡	12.9 ± 1.1 †
MRTG	2.8 ± 0.5	1.4 ± 0.9 *	2.6 ± 0.4 †	1.4 ± 0.6 *‡	3.2 ± 0.5 †
TTG	176 ± 13	123 ± 65	178 ± 9	119 ± 52	191 ± 30 †
*Crotalus organus abyssus* (0.5 µg/mL)					
TMRTG	11.7 ± 1.2	23.2 ± 5.3 *	14.6 ± 2.7 †	13.7 ± 3.3 †	21.2 ± 5.5
MRTG	3.6 ± 0.6	0.6 ± 0.6 *	3.2 ± 0.6 †	2.9 ± 0.6 *†	0.6 ± 0.6
TTG	203 ± 10	72 ± 79 *	196 ± 17 †	188 ± 15 †	68 ± 57
*Porthidium nasutum* (1.0 µg/mL)					
TMRTG	11.8 ± 0.7	25.4 ± 6.4 *	12.9 ± 3.1 †	22.8 ± 4.1 *‡	29.0 ± 1.5
MRTG	2.8 ± 0.5	0.6 ± 0.9 *	3.4 ± 1.7 †	1.0 ± 0.7 *‡	0.1 ± 0.1
TTG	176 ± 13	57 ± 87 *	187 ± 37 †	107 ± 68	5 ± 9
*Sistrurus catenatus edwardsii* (0.5 µg/mL)					
TMRTG	11.7 ± 1.2	21.0 ± 7.7 *	15.0 ± 5.3 *	26.7 ± 4.6 *‡	25.9 ± 6.3
MRTG	3.6 ± 0.6	0.6 ± 0.6 *	2.8 ± 1.0 *†	0.1 ± 0.1 *‡	0.1 ± 0.1
TTG	203 ± 10	59 ± 68 *	186 ± 17 †	2 ± 4 *†‡	3 ± 4
*Sistrurus miliarius barbourii* (0.25 µg/mL)					
TMRTG	11.7 ± 1.2	26.6 ± 3.9 *	13.8 ± 3.2 †	24.0 ± 5.8 *‡	29.1 ± 2.3
MRTG	3.6 ± 0.6	0.5 ± 0.5 *	3.2 ± 0.9 †	0.6 ± 0.4 *‡	0.0 ± 0.0
TTG	203 ± 10	42 ± 67 *	210 ± 19 †	66 ± 69 *‡	1 ± 2

Data are presented as mean ± SD. Control = no venom or other additions; V = venom addition without exposure to other compounds; V/CO = venom addition after exposure to tricarbonyldichlororuthenium (II) dimer (CORM-2); V/inactive releasing molecule (iRM) = venom addition after exposure to inactivated CORM-2; V/PHA = venom addition after exposure to PHA. All venoms were exposed in isolation to 100 µM CORM-2 and iRM. The final isolated venom exposure PHA concentration was 30 mM for all venoms. All venom mixtures were 1% (*v*/*v*) additions to plasma mixtures. TMRTG = time to maximum rate of thrombus generation (min); MRTG = maximum rate of thrombus generation (dynes/cm^2^/sec); TTG = total thrombus generation (dynes/cm^2^). * *p* < 0.05 vs. Control, † *p* < 0.05 vs. V, ‡ *p* < 0.05 vs. V/CO.

**Table 5 toxins-11-00094-t005:** Coagulation kinetics of procoagulant venoms ± CO or ± PHA in plasma.

Parameter	V	V/CO	V/iRM	V/PHA
*Agkistrodon bilineatus* (5 µg/mL)				
TMRTG	6.3 ± 0.4	12.9 ± 1.0 *	6.6 ± 0.3 †	14.9 ± 0.2 *
MRTG	2.7 ± 0.1	3.6 ± 0.6 *	2.5 ± 0.2 †	0.0 ± 0.0 *
TTG	144 ± 4	136 ± 37	143 ± 7	0 ± 0 *
*Atropoides olmec* (10 µg/mL)				
TMRTG	13.5 ± 1.1	12.6 ± 0.6	13.8 ± 0.9	13.2 ± 1.5
MRTG	0.8 ± 0.4	2.5 ± 0.3 *	0.7 ± 0.2 †	0.9 ± 0.5
TTG	51 ± 25	121 ± 15 *	45 ± 13 †	50 ± 30
*Bothrops colombiensis* (2 µg/mL)				
TMRTG	1.9 ± 0.1	5.0 ± 0.2 *	2.1 ± 0.2 †	10.6 ± 1.5 *
MRTG	9.5 ± 1.0	3.3 ± 0.5 *	8.7 ± 1.3 †	0.1 ± 0.0 *
TTG	207 ± 10	143 ± 17 *	208 ± 15 †	2 ± 0 *
*Bothrops jararaca, S* (2 µg/mL)				
TMRTG	1.0 ± 0.1	2.3 ± 0.2 *	1.3 ± 0.2 *†	9.2 ± 4.3 *
MRTG	12.0 ± 0.8	9.6 ± 1.5 *	12.0 ± 0.9	2.8 ± 2.6 *
TTG	178 ± 9	196 ± 19	195 ± 9	81 ± 63 *
*Bothrops jararaca, SE* (2 µg/mL)				
TMRTG	1.6 ± 0.1	3.9 ± 0.4 *	1.7 ± 0.1 †	10.3 ± 1.1 *
MRTG	11.8 ± 2.1	6.2 ± 1.2 *	11.6 ± 1.4 †	4.6 ± 0.8 *
TTG	204 ± 32	175 ± 24	197 ± 18	158 ± 10 *
*Bothrops moojeni* (2 µg/mL)				
TMRTG	1.2 ± 0.1	3.7 ± 0.7 *	1.2 ± 0.0 †	11.2 ± 0.6 *
MRTG	11.2 ± 1.5	6.5 ± 1.5 *	10.8 ± 1.5 †	1.0 ± 0.2 *
TTG	167 ± 27	179 ± 23	153 ± 17	62 ± 16 *
*Crotalus simus tzabcan* (3 µg/mL)				
TMRTG	6.0 ± 1.5	15.1 ± 2.2 *	12.3 ± 7.9 *	10.4 ± 2.5 *
MRTG	0.1 ± 0.0	2.3 ± 0.5 *	0.1 ± 0.1 †	0.1 ± 0.0
TTG	7 ± 1	155 ± 31 *	6 ± 3 †	93 ± 19 †

Data are presented as mean ± SD. V = venom addition without exposure to other compounds; V/CO = venom addition after exposure to CORM-2; V/iRM = venom addition after exposure to inactivated CORM-2; V/PHA = venom addition after exposure to PHA. *Bothrops moojeni* and *Crotalus simus tzabcan* venoms were exposed in isolation to 1 mM CORM-2 and iRM; the other venoms were exposed in isolation to 100 µM CORM-2 and iRM. The final isolated venom exposure PHA concentration was 30 mM for all venoms. All venom mixtures were 1% (*v*/*v*) additions to plasma mixtures. TMRTG = time to maximum rate of thrombus generation (min); MRTG = maximum rate of thrombus generation (dynes/cm^2^/sec); TTG = total thrombus generation (dynes/cm^2^). * *p* < 0.05 vs. V, † *p* < 0.05 vs. V/CO.

**Table 6 toxins-11-00094-t006:** Kinetomic profiles of Pan-American Viper Venoms.

Species	Activity	[µg/mL]	CO	PHA
**North American Snakes**				
*Agkistrodon contortrix contortrix*	A	10	−	NT
*Agkistrodon contortrix laticinctus*	A	30	+	NT
*Agkistrodon contortrix pictigaster*	A	11	NT	NT
*Agkistrodon contortrix mokasen*	A	8	+	NT
*Agkistrodon piscivorus leucostoma*	A	5	+	NT
*Agkistrodon piscivorus piscivorus*	A	5	+	NT
*Crotalus adamanteus*	TLA	5	+	NT
*Crotalus atrox*	A	2	+	NT
*Crotalus cerastes cercobombus*	A	2	+	+
*Crotalus horridus horridus*	TLA/TGA	5	+	NT
*Crotalus organus abyssus*	A	0.5	?	−
*Crotalus oreganus cerberus*	A	2	+	−
*Crotalus oreganus helleri*	TLA	10	NT	NT
*Crotalus oreganus oreganus*	A	2	+	NT
*Crotalus ruber ruber*	A	10	NT	NT
*Crotalus viridis viridis*	A	10	NT	NT
*Sistrurus catenatus edwardsii*	A	0.5	+	−
*Sistrurus miliarius barbourii*	A	0.25	+	−
**Central American Snakes**				
*Agkistrodon bilineatus*	TGA	5	+	+
*Atropoides olmec*	TLA	10	+	−
*Bothriechis schlegelii*	A	0.15	?	−
*Crotalus basiliscus*	A	1	+	−
*Crotalus simus simus*	TLA	2	NT	NT
*Crotalus simus tzabcan*	TLA	2	+	+
*Porthidium nasutum*	A	1	+	−
**South American Snakes**				
*Bothrops asper*	TGA	2	+	NT
*Bothrops atrox*	TGA	20	NT	NT
*Bothrops colombiensis*	TGA	2	+	+
*Bothrops jararaca*	TGA	2	+	+
*Bothrops moojeni*	TGA	2	+	+
*Bothrops neuwiedi*	TGA	20	NT	NT
*Lachesis muta muta*	A/TLA	2	+	NT

Activity: A = anticoagulant; TLA = thrombin-like activity; TGA = thrombin-generating activity. Heme modulation: CO = affected by carbon monoxide; PHA = affected by PHA; + = inhibited by the indicated molecule; − = not affected by the indicated molecule; NT = not tested; ? = both CORM-2 and iRM inhibited this venom.

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
