# Peer review of "De Novo Assessment and Review of Pan-American Pit Viper Anticoagulant and Procoagulant Venom Activities via Kinetomic Analyses"

_toxins, 2019, doi:10.3390/toxins11020094_

Round 1

Reviewer 1 Report

The authors modified their manuscript which had been rejected, but I dont see any new arguments which would convince me to reconsider my initial judgement. They claim that a heme group is associated with snake venom coagulant and anticoagulant factors. In a publication of Suntravat et al. (ref. 12) a Crotalus metalloproteinase is described which is considered to contain a heme-molecule. However, it is not clear whether this molecule is part of the structure of the enzyme or is attached by non-covalent binding. This is the genral problem in studies using crude venoms which contain iron as trace compounds, which may certainly bind CO released from CORM-2. In a previous paper (ref. 5) even CO-inhibition of haemotoxic effects were observed in cobra venom, but phospholipases A2 definitely do not contain heme-groups which may explain any inhibitory activity seen. In a previous paper, the authors have also shown  that addition of iron and CORM-2 to a venom solution affects its coagulant activity. This rather seems to indicate that inhibitory effects are due to some still unknown mechanisms such as interaction of the respective enzyme with iron (not bound to the enzyme) and bound CO. Studies as presented in the present paper using a number of crude venoms  and testing the pro- and a coagulant inhibition are not helpful to explain this effect assayed by thrombelastography, a well established method, providing some data of venom activities, but it is not useful to explain inihibitory effects as seen in vitro. 

Of course, the use of CO-releasing compounds has been described in numerous publications, but its potential clinical use seems to me rather doubtful (you may believe or not data of studies from China). Snake envenoming studies have already shown that application of any agent preventing spreading or inactivating venom factors needs to be done within a short  period of time and are already too late in such an event. To "render the enzymes quiescent prior to antivenom administration" is a theoretical, but not a practical option in snakebite treatment.

Author Response

Reviewer 1

“The authors modified their manuscript which had been rejected, but I dont see any new arguments which would convince me to reconsider my initial judgement. They claim that a heme group is associated with snake venom coagulant and anticoagulant factors. In a publication of Suntravat et al. (ref. 12) a Crotalus metalloproteinase is described which is considered to contain a heme-molecule. However, it is not clear whether this molecule is part of the structure of the enzyme or is attached by non-covalent binding.”

We appreciate this point of view and will now try again to convince the reviewer of the validity of our approach.  We now include a new paragraph in Discussion that directly addresses why heme modulating agents exert their effects only if a heme group is attached to the enzymes investigated.  As a rule, heme bound enzymes have the heme group placed as a post translational event, so the published structures of all heme bound enzymes would not be expected to include the group as part of the primary sequence of the molecule.  Lastly, heme groups are bound to proteins a variety of ways, but primarily via a critical histidine moiety within the protein structure proper.

“This is the genral problem in studies using crude venoms which contain iron as trace compounds, which may certainly bind CO released from CORM-2.” 

Biochemically, this is not the case.  First of all, free iron in aerobic solution is rapidly oxidized to the Fe+3 state, which does not bind with CO.  Second, free iron would rapidly react with whatever anion was near it, and would not function as a modulatory entity.  Similarly, free hemoglobin (which is what we assume the reviewer means) released into the venom from the trauma of milking the snake would be very small in concentration and would not be expected to somehow rapidly bind to the key enzymes without some sort of intermediary step, such as that associated with post translational modifications.

“In a previous paper (ref. 5) even CO-inhibition of haemotoxic effects were observed in cobra venom, but phospholipases A2 definitely do not contain heme-groups which may explain any inhibitory activity seen.”

Again, the author has no evidence that PLA2 do not have heme groups attached in a post translational manner.  In a future manuscript with more cobras, we will make this point again.  Reference 32 demonstrates a sigmoidal inhibitory kinetic response of an isolated PLA2 in response to CO, the sine quo non of heme modulation.

“In a previous paper, the authors have also shown that addition of iron and CORM-2 to a venom solution affects its coagulant activity. This rather seems to indicate that inhibitory effects are due to some still unknown mechanisms such as interaction of the respective enzyme with iron (not bound to the enzyme) and bound CO.”

Actually, we demonstrated in the manuscript the author cites that while the combination of iron and CORM-2 affected activity, a solution of iron alone did not.  Thus, iron was just a bystander, not affecting venom activity.  This is our reference #6, page 1975, table 2, towards the bottom.

“Studies as presented in the present paper using a number of crude venoms and testing the pro- and a coagulant inhibition are not helpful to explain this effect assayed by thrombelastography, a well established method, providing some data of venom activities, but it is not useful to explain inihibitory effects as seen in vitro.”

I am puzzled by this comment.  As the reviewer knows, I am an accepted international expert on this methodology.  The better part of my career over the past 20 years has been spent on using thrombelastographic techniques to identify pathological states, key enzyme systems at play in those states, and the inhibition of the enzyme activities based on coagulation and fibrinolysis.  If I can create a control set of coagulation kinetic values, then I can establish a change in these values with venom, purified enzyme, or anything else that is bothering the various coagulation cascades.  If I expose the analyte of interest in isolation to an inhibitor that should bind to it, and then place it in plasma and see a reversal of the enzymatic effects of the analyte, then I should be able to comment on the inhibition of the enzymes studied.  I have published dozens of manuscripts in several disciplines using this accepted scientific approach.  Lastly, Dr. Bryan Fry’s group has been using thrombelastography over the past year to demonstrate that various antivenoms can neutralize the in vitro coagulopathies caused by literally dozens of crude venoms.  In summary, we must disagree with the reviewer on this point, given the preponderance of the evidence.

“Of course, the use of CO-releasing compounds has been described in numerous publications, but its potential clinical use seems to me rather doubtful (you may believe or not data of studies from China).”

This question remains to be answered, and there is ongoing development of new CORMs every few months it seems.  It is hard for us to know what studies the reviewer is alluding to from China.  It has already been demonstrated in vivo in rabbits (a higher mammal) that CO exposed Crotalus atrox venom is inhibited by me in reference #49.  Only future investigation will allay the reviewer’s doubts, but I think we can all agree that this is beyond the scope of the present work.

“Snake envenoming studies have already shown that application of any agent preventing spreading or inactivating venom factors needs to be done within a short  period of time and are already too late in such an event. To "render the enzymes quiescent prior to antivenom administration" is a theoretical, but not a practical option in snakebite treatment.”

The same argument could be applied to antivenom, as the “fang-to-needle” time predicts the success of antivenom administration – yet, it is routine to administer antivenom hours after envenomation.  Given the long shelf-life of CORMs and potential to administer a solution of them within minutes emergently by the victim or first responders, the effects of CO mediated inhibition may be very important but critically remain to be proven in future investigations.  As the reviewer knows, there is no thing such as a “standardized” snake bite.  Different locations, amounts of venom, type of venom, etc. will always affect clinical outcome.  Further, there is a depot effect wherein it has been observed that after the first round of antivenom is cleared from the circulation, the bite victim can exhibit symptoms of envenomation.  This is likely secondary to release of venom from the edematous extremity bitten, which clearly contained active venom that could be released into the circulation days to weeks after envenomation.  Somewhere in this spectrum an acute (minutes to hours) local administration of CORMs in solution or biogel form would be expected to inactivate venom activity within the spread of CO diffusion.  As the reviewer knows, CO is one of the most rapidly diffusing molecules known, so if an ongoing release of CO from longer acting CORMs follows the initial burst from short acting CORMs, it is entirely possible that this therapy may be very effective – especially to local tissue damage.  Again, these issues are debatable and remain to be proven empirically one way or the other.  Thus, they are beyond the scope of our manuscript.

Reviewer 2 Report

It is interesting to have classified snake venom from a point of view of blood coagulation pattern. And it may be important in taxonomy to have shown the kinetomic profiles of Pan-American viper venoms.

  An experimental design of this study does not have the major problem, and the results that authors aim at are shown properly. However, analysis of the snake venom by this technique does not have the novelty so that authors show it with a lot of references cited.

Also, the authors stated that local application of CO via releasing molecules may serve as an adjunct therapy prior to antivenom administration, but the clinical application may be difficult because there is a problem of the toxicity for the blood of CO. Therefore, it is desirable to be described more carefully about the clinical application of these results in the therapeutic treatment.

Author Response

Reviewer 2

“It is interesting to have classified snake venom from a point of view of blood coagulation pattern. And it may be important in taxonomy to have shown the kinetomic profiles of Pan-American viper venoms.”

We appreciate the reviewer’s kind comments.

“An experimental design of this study does not have the major problem, and the results that authors aim at are shown properly. However, analysis of the snake venom by this technique does not have the novelty so that authors show it with a lot of references cited.”

We appreciate this comment.  This manuscript was intended to document the diversity and novelty of the kinetomic profiles of Pan-American viper venoms with methods developed by us and already validated with other venoms from snakes around the world.  Nevertheless, while we have published a number of works in the recent past, we posit that the methodology is still relatively new and novel, providing new insights into each group of viper venoms we investigate.

“Also, the authors stated that local application of CO via releasing molecules may serve as an adjunct therapy prior to antivenom administration, but the clinical application may be difficult because there is a problem of the toxicity for the blood of CO.  Therefore, it is desirable to be described more carefully about the clinical application of these results in the therapeutic treatment.”

We understand the apprehension the reviewer expresses concerning the potential for CO poisoning systemically.  The reality is that several-fold administration of CORM-2 that would be delivered locally would result in far less of a change in CO concentration in the systemic circulation than has already been demonstrated to be innocuous in rabbit models.  We now include a new paragraph in Discussion to outline this matter in greater detail.

Reviewer 3 Report

In  this review article the authors use thromboelastographic coagulation kinetics for analysis of several snake venoms in order to distinguish whether the main effect of these toxic venoms result from being either predominantly an anticoagulant or a procoagulant action in their envenoming profile. Furthermore they studied the possible thought action of CO or O-phenylhidroxylamine (PHA) as inhibitors of the hemotoxic venom effects. The hypothesis supporting the use of CO or PHA is based on their previous finding that several enzymes involved in the determination of the final venom action are modulated by heme group bound to them and thus being inhibitable by these  compounds. Combining the data obtained from these experiments the authors designed a so named venom kinetomic (a term coined by their own) where the pattern thus originated allow to distinguish both kind of venom activities and thus a classification of snakes according to these two types of action.

The experiments were well conducted and sounding results were obtained by applying these parameters to the venom of 17 snakes from different regions. However, a few points must be considered by the authors before  the acceptance of the manuscript for publication in TOXINS. 

Major concern: Very high concentrations of CO were used to obtain inhibitory effects on the venoms (page 8, lines 221-223). In fact 70 uM and in some cases 10 to 15-folds higher concentration of CO are several folds the ratio E:I proportion. Wouldn’t this condition be too artificial in order to allow significant interpretation to support the hypothesis of CO inhibitory effect? What about the concentrations of PHA?

Minor concerns:

1)   There are many abbreviations used throughout the text. It is then recommended that an Abbreviation List should be included in an appropriate place;

2)   The manuscript requires an English revision. Some examples:  “In order to further this undertaking of documenting …” (page 2, line 47);

“… kinetic profiles that are identical that are generated…” (page 2, lines 66-67); the expression (in several places) “venom exposed plasma” could be better reading as “venom-exposed-plasma; “a unique circumstance” (page 5. Line 150); “…venom exposure to CO exposure …” (page 5, lines 153 -155);

3)   Last sentence in page 3 (lines 109-111): Bothriechis schlegelli is not also an exception?

4)   Page 6, lines 156-157: the expression refers to which venom?

5)   Page 7, lines 191-192: It seems that the exceptions mentioned should include Agkistrodon contortrix laticinctus;

Author Response

Reviewer 3

“In  this review article the authors use thromboelastographic coagulation kinetics for analysis of several snake venoms in order to distinguish whether the main effect of these toxic venoms result from being either predominantly an anticoagulant or a procoagulant action in their envenoming profile. Furthermore they studied the possible thought action of CO or O-phenylhidroxylamine (PHA) as inhibitors of the hemotoxic venom effects. The hypothesis supporting the use of CO or PHA is based on their previous finding that several enzymes involved in the determination of the final venom action are modulated by heme group bound to them and thus being inhibitable by these  compounds. Combining the data obtained from these experiments the authors designed a so named venom kinetomic (a term coined by their own) where the pattern thus originated allow to distinguish both kind of venom activities and thus a classification of snakes according to these two types of action.”

We appreciate this succinct summary of our work, which is a combination of newly derived experimental data and a review of works past.

“The experiments were well conducted and sounding results were obtained by applying these parameters to the venom of 17 snakes from different regions. However, a few points must be considered by the authors before the acceptance of the manuscript for publication in TOXINS.”

We appreciate this comment as well.

“Major concern: Very high concentrations of CO were used to obtain inhibitory effects on the venoms (page 8, lines 221-223). In fact 70 uM and in some cases 10 to 15-folds higher concentration of CO are several folds the ratio E:I proportion. Wouldn’t this condition be too artificial in order to allow significant interpretation to support the hypothesis of CO inhibitory effect? What about the concentrations of PHA?”

As we point out in the new paragraph in Discussion concerning heme biology and CO reactivity, CO is not a radical and effects biological processes by binding to the ferrous iron within heme attached to proteins.  Further, it is not known how many heme groups are attached to any particular protein, how many hemes must be in the carboxy state to change a protein’s kinetic behavior, how many compounds in a sample are heme bearing, and what ratio of moles of CO to moles of heme bearing enzyme are needed to quickly and maximally change enzyme behavior.  In order to maximize kinetic response in any system, there is always an excess of inhibitor/activator to active site.  Lastly, we are not sure what the reviewer means by artificial – the entire diagnostic system we are using can easily be accused of not being particularly natural.  It was designed a priori to discern the predominant enzymatic species responsible for changes in coagulation and subsequently to assess if heme modulation would affect this enzymatic activity. 

The interpretation of the effects of CO on any system with CORMs is based on a change observed with a CORM not being observed with an inactivated CORM.  Concentration has little to do with it as far as we can tell from the available in vitro literature using the compounds in the ranges we have in this and other recent works.  We have generated sigmoidal shaped CO concentration-enzyme inhibition curves with CO concentrations from 0 to 700 micromolar depending on the species involved.  This implies heme-based biology at play, with the target enzyme either more or less vulnerable to CO modulation.

As for the concentrations of PHA, we have used these for several years in many different biological samples, and it either has an effect or not based on likely heme-based biology.  We have seen PHA decrease or do nothing to venom activity in all our investigations to date.

Minor concerns:

1)   “There are many abbreviations used throughout the text. It is then recommended that an Abbreviation List should be included in an appropriate place;”  The addition of a list in Results has been done to accommodate this request.

2)   “The manuscript requires an English revision. Some examples:  “In order to further this undertaking of documenting …” (page 2, line 47);

“… kinetic profiles that are identical that are generated…” (page 2, lines 66-67); the expression (in several places) “venom exposed plasma” could be better reading as “venom-exposed-plasma; “a unique circumstance” (page 5. Line 150); “…venom exposure to CO exposure …” (page 5, lines 153 -155);”  We have modified the text as suggested with the exception of the double hyphenated phrase as it is cumbersome and distracts from the associated text.

3)   “Last sentence in page 3 (lines 109-111): Bothriechis schlegelli is not also an exception?”  Yes, and thank you for finding this error.

4)   “Page 6, lines 156-157: the expression refers to which venom?”  Atropoides olmec, which is now indicated.

5)   “Page 7, lines 191-192: It seems that the exceptions mentioned should include Agkistrodon contortrix laticinctus;”  No, for as table 5 indicates, CO inhibits the activity of this venom.